# Physical Exercise Decreases Endoplasmic Reticulum Stress in Central and Peripheral Tissues of Rodents: A Systematic Review

Matheus Santos de Sousa Fernandes [1], Georgian Badicu [2,*], Gabriela Carvalho Jurema Santos [3], Tayrine Ordonio Filgueira [4], Rafael dos Santos Henrique [5], Raphael Fabrício de Souza [6], Felipe J. Aidar [6], Fabrício Oliveira Souto [4], Patrícia Chakur Brum [7] and Claudia Jacques Lagranha [1]

[1] Graduate Program in Neuropsychiatry and Behavioral Sciences, Center for Medical Sciences, Federal University of Pernambuco, Recife 507400-600, Pernambuco, Brazil; theusfernandes10@hotmail.com (M.S.d.S.F.); claudia.lagranha@ufpe.br (C.J.L.)

[2] Department of Physical Education and Special Motricity, Transilvania University of Brasov, 500068 Brasov, Romania

[3] Postgraduate Program in Nutrition, Federal University of Pernambuco, Recife 507400-600, Pernambuco, Brazil; gaby9carvalho@gmail.com

[4] Graduate Program in Applied Health Biology, Keizo Asami Immunopathology Laboratory, Federal University of Pernambuco, Recife 507400-600, Pernambuco, Brazil; tayrine.ordonio@ufpe.br (T.O.F.); fabicio.souto@ufpe.br (F.O.S.)

[5] Department of Physical Education, Federal University of Pernambuco, Recife 507400-600, Pernambuco, Brazil; rafael.sherique@ufpe.br

[6] Department of Physical Education, Federal University of Sergipe, São Cristovão 49100-000, Sergipe, Brazil; raphaelctba20@hotmail.com (R.F.d.S.); fjaidar@gmail.com (F.J.A.)

[7] School of Physical Education and Sport, Universidade de São Paulo, São Paulo 05508-900, São Paulo, Brazil; pcbrum@usp.br

* Correspondence: georgian.badicu@unitbv.ro

**Abstract:** Endoplasmic reticulum stress (ER stress) affects many tissues and contributes to the development and severity of chronic diseases. In contrast, regular physical exercise (PE) has been considered a powerful tool to prevent and control several chronic diseases. The present systematic review aimed to evaluate the impact of different PE protocols on ER stress markers in central and peripheral tissues in rodents. The eligibility criteria were based on PICOS (population: rodents; intervention: physical exercise/physical training; control: animals that did not undergo training; outcomes: endoplasmic reticulum stress; studies: experimental). The PubMed/Medline, Science Direct, Scopus, and Scielo databases were analyzed systematically. Quality assessment was performed using SYRCLE's risk of bias tool for animal studies. The results were qualitatively synthesized. Initially, we obtained a total of 2.490 articles. After excluding duplicates, 30 studies were considered eligible. Sixteen studies were excluded for not meeting the eligibility criteria. Therefore, 14 articles were included. The PE protocol showed decreased levels/expression of markers of ER stress in the central and peripheral tissues of rodents. PE can decrease ER stress by reducing cellular stress in the cardiac, brain, and skeletal muscle tissues in rodents. However, robust PE protocols must be considered, including frequency, duration, and intensity, to optimize the PE benefits of counteracting ER stress and its associated conditions.

**Keywords:** exercise; physical activity; organelle; cellular stress; metabolism

## 1. Introduction

Cytoplasmic organelles contribute to the maintenance of cellular homeostasis of eukaryotes, including the endoplasmic reticulum (ER) [1]. ER is formed from plasma membrane invaginations and is involved in several functions, including the biosynthesis of lipids and proteins and their storage, as well as regulating enzyme activity and metabolism [2,3].

Under conditions of cellular stress (e.g., malnutrition, overnutrition, and physical inactivity), ER induces an increase in the level of unfolded proteins [4–6]. The accumulation of unfolded proteins in ER lumen is responsible for triggering an unfolded protein response (UPR), which promotes morphophysiological changes capable of generating ER stress [7].

At the molecular cellular level, ER stress results in an increased release of Glucose-Regulated Protein 78 (GRP78), which is associated with other ER membrane-adhered cell signaling factors, including protein kinase RNA (PKR)-like ER kinase (PERK) and Activating Transcription Factor-6 (ATF-6). Altogether, these proteins modulate protein folding, maturation, transport, and degradation, culminating in the activation of c-Jun N-terminal Kinase (JNK) and apoptosis [8]. As a consequence, PERK phosphorylates Eukaryotic Initiation Factor 2$\alpha$ (eIF2$\alpha$), decreasing the translation of messenger RNA and the loading of proteins to the ER. In addition, Activating Transcription Factor 4 (ATF-4) translation is activated by the PERK-eIF2$\alpha$ axis, initiating the transcription of apoptotic factors, such as homologous protein C (CHOP), upregulated by the ATF-6 pathway [8,9].

ER stress is also associated with the dysfunction of central tissues and peripheral tissues, including the heart, brain, and skeletal muscle [10–12], which ultimately leads to type 2 diabetes mellitus, obesity, cardiovascular disease, depression, and other neurodegenerative conditions [13–17]. However, there is no complete understanding of the different mechanisms that promote ER stress, as well as the lack of pharmacological strategies to counteract it [18]. Therefore, the use of lifestyle-related interventions, such as PE, appears as an alternative in the prevention and treatment of ER stress. Several systematic reviews with meta-analysis demonstrate the effectiveness of different PE protocols, including aerobic (AE) and resistance (RE) exercise, in promoting health, increasing physical capacity, as well as preventing and neutralizing different types of cellular stress, including oxidative stress and mitochondrial dysfunction [19–21]. However, the role of PE on different ER stress markers is not entirely clear. Furthermore, there is still no systematic review in the scientific literature that investigated this relationship between PE and ER stress in different tissues. Therefore, the objective of this systematic review is to evaluate the impacts of different PE protocols on different ER stress markers in central and peripheral tissues in rodent models.

## 2. Materials and Methods

This is a systematic literature review, developed to answer the following research question: "What are the responses promoted by PE in ER stress markers in central and peripheral tissues of rodents". The process of conducting the review was carried out based on the Preferred Reporting Items for Systematic Reviews and Meta-Analyzes (PRISMA) [22] guidelines and registered in PROSPERO CRD42022299710.

### 2.1. Eligibility Criteria

The PICOS of the current study was as follows: population: rodents; intervention: physical exercise/physical training; control: animals that did not undergo training; outcomes: endoplasmic reticulum stress; studies: experimental. Based on the PICOS, Table 1 presents the inclusion and exclusion criteria.

**Table 1.** Eligibility criteria based on the PICOS strategy.

|  | Inclusion Criteria | Exclusion Criteria |
|---|---|---|
| Population | Rodents | Humans and other species |
| Intervention | Physical exercise/physical training | Absence of the physical exercise/physical training |
| Comparator | Animals that did not undergo training | Absence of the control group |
| Outcomes | Endoplasmic reticulum stress markers | No endoplasmic reticulum stress markers |
| Study design | Experimental | Observational, reviews, case reports, and scientific abstracts |

### 2.2. Information Sources and Search Strategy

The bibliographic search was performed in the National Library of Medicine/Medical Literature Analysis and Retrieval System (PubMed/Medline), Scopus, Scientific Electronic Library Online (Scielo), and ScienceDirect databases. The following terms were used to search for articles (Exercise) OR (Physical Activity) OR (Physical Exercise) OR (Physical Exercises) OR (Exercise Training) OR (Exercise Trainings) AND (Endoplasmic Reticulum Stress) OR (Endoplasmic Reticulum Stresses) OR (Stress, Endoplasmic Reticulum) OR (Stresses, Endoplasmic Reticulum). Modifications in the search strategy were carried out as needed by the database (Table 2). No limits were set on the publication date and language. Duplicate articles were removed with the help of ENDNOTE X20 software. This step of the review was conducted by two independent researchers, with any discrepancies resolved by a third reviewer.

**Table 2.** Sample search strategy on databases.

| Database | Code Line |
|---|---|
| Pubmed/Medline, Scopus, and Scielo | (Exercise) OR (Physical Activity) OR (Physical Exercise) OR (Physical Exercises) OR (Exercise Training) OR (Exercise Trainings) AND (Endoplasmic Reticulum Stress) OR (Endoplasmic Reticulum Stresses) OR (Stress, Endoplasmic Reticulum) OR (Stresses, Endoplasmic Reticulum) |
| ScienceDirect | ("Exercise") OR ("Physical Activity") OR ("Physical Exercise") AND ("Endoplasmic Reticulum Stress") OR ("Endoplasmic Reticulum Stresses") OR ("Stress, Endoplasmic Reticulum") OR ("Stresses, Endoplasmic Reticulum") |

### 2.3. Selection

This step of the review was carried out by two independent investigators (MSSF and GCJS) according to the predefined criteria in the protocol. Discrepancies were resolved by a third evaluator. Initially, the titles and abstracts were evaluated. Those who did not meet the review criteria were excluded.

### 2.4. Data Extraction and Items Collection

For the included articles, the following data were collected: (1) author; (2) year of publication; (3) rodent species; (4) sex; (5) age (weeks); (6) number of animals per study; (7) type of physical exercise; (8) and protocol characteristics, including frequency, duration, and intensity. We then collected the key outcomes related to the central brain (prefrontal cortex, cortex, hypothalamus, and hippocampus), heart (myocardium and left ventricle), and peripheral tissues, including different skeletal muscles (anterior tibial, EDL, gastrocnemius, quadriceps, and soleus). Finally, we collected endoplasmic reticulum stress markers.

### 2.5. Methodological Quality Assessment

The assessment of the methodological quality of the studies was performed using SYRCLE's risk of bias tool for animal studies [23], a derivative of the Cochrane Collaboration's risk of bias (RoB) tool that evaluates clinical trials. The SRYCLE strategy has validity and reproducibility. SYRCLE includes ten questions to assess the performance, friction, and bias of scientific articles included in the studies; the answers were scored with "Yes", indicating a low risk of bias, or "No", indicating a high risk of bias. Articles were evaluated as low, acceptable, or high quality. Those assessed as of low quality were excluded from the review. This assessment was performed by two independent researchers (MSSF and GCJS), with any discrepancies resolved by a third assessor (TOF).

### 3. Results

The flowchart with the articles that were included in this systematic review is shown in Figure 1. Initially, we obtained a total of 2490 articles in four databases (PubMed/Medline: 226; Scopus: 349; Science Direct: 1914; Scielo: 1). After excluding duplicates (n = 242),

30 studies were considered eligible and further analyzed. Sixteen studies were excluded for not fully meeting the eligibility criteria. At the end of the process, 14 articles were included.

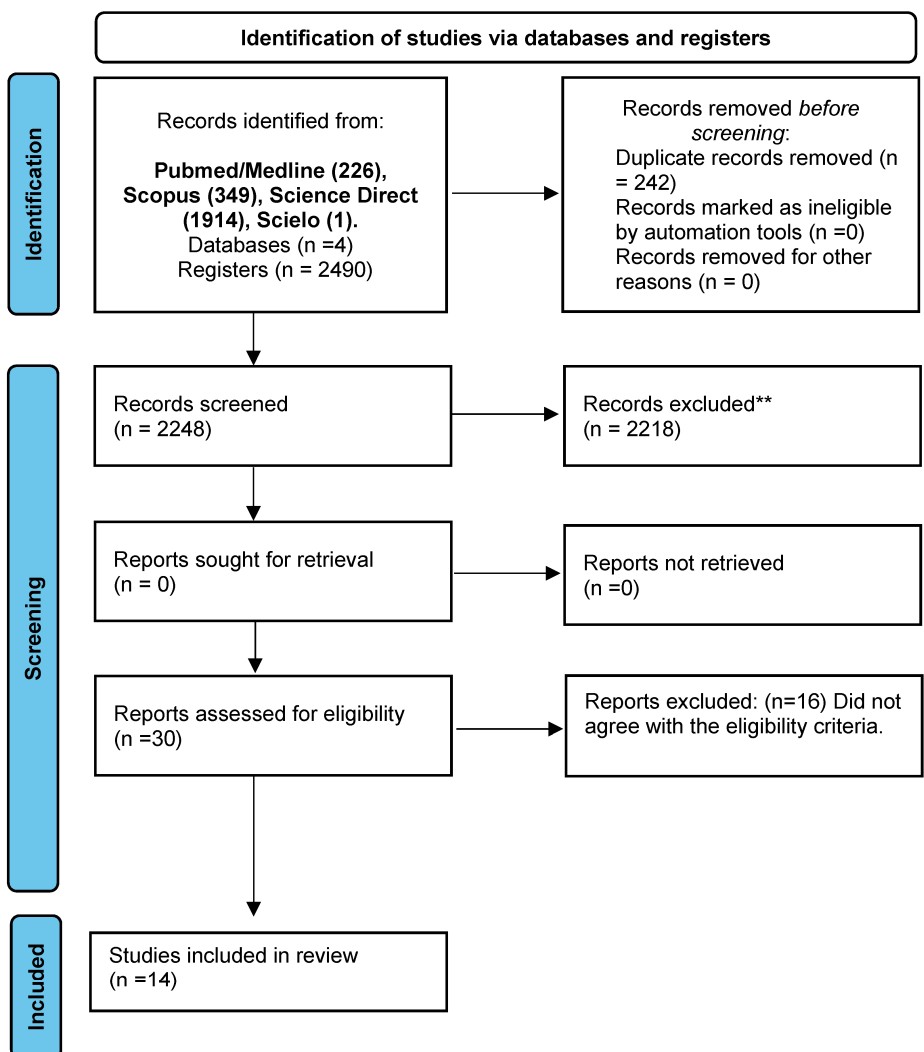

**Figure 1.** PRISMA 2020 flow diagram for new systematic reviews, which included searches of databases and registers only. ** $p \leq 0.01$.

Table 3 presents the main characteristics of the included studies. The articles included were published between 2007 and 2021, and the sample size ranged between 12 and 150 rodents. Eleven studies used male rodents; one study only used both sexes and in two studies, the sex was not informed. A wide variety of rodent species was found in the included studies: nine studies used C57BL6 mice; four studies used Sprague Dawley rats and one study used Institute of Cancer Research (ICR) mice. The age of the rodents ranged from 5 weeks to 20 months. Several PE protocols were used, including twelve studies that used aerobic exercise as an intervention; one study only applied resistance exercise and in one study, only both modalities were used. In eight studies, the frequency of 5 days/week was applied; the duration of the protocols and the session varied from 1 to 21 weeks and between 10 to 60 min, respectively. In establishing intensity, five studies used speed in meters/minutes; four studies used exercise to exhaustion; one study only used maximal oxygen volume (VO$_{2\text{Max}}$); one study used the percentage of body weight of rodents in resistance exercise and VO$_{2\text{Max}}$ for aerobic exercise, and in three studies, the intensity measurement was not included because it was voluntary PE in a running wheel.

Among the studies included, six studies assessed cardiac tissue, four studies focused on the myocardium [24–27], and two were on the left ventricle [28,29]. As shown in Table 2,

14 ER stress markers were evaluated after PE, including ATF3, ATF4, BIP, CHOP, p-eIF2α, eIF2α, GRP78, HSP72, p-IRE, IRE, JNK, p-PERK, PERK, and PKG, as well as the expression of PDE5, an ER stress regulator. A significant decrease in ER stress markers, including ATF4, CHOP, GRP78, p-PERK, PERK, and PKG, was observed after the AE and swimming. Similarly, after the RE protocol, there was also a significant reduction in the levels of p-PERK and PERK.

However, no significant differences were observed in the levels of CHOP and GRP78. Finally, AE increased the BIP, HSP72, and PDE5 levels (Table 4).

Within the included studies, different areas of the brain were evaluated, being one study performed on the hypothalamus [30], one study on the prefrontal cortex [31], and one study analyzed three areas: the cortex, hippocampus, and hypothalamus [32]. In the study that evaluated only the hypothalamus (Table 3), the AE was performed for 8 weeks on a treadmill, with and without an upward inclination. After the intervention, there was an increase in BIP, GRP94, p-IRE1, and IRE1 expression. There were no differences in the ATF6 levels. However, when using the downward inclination protocol, there was an increase in ATF6, BIP, GRP94, p-IRE1, and IRE1 expression. In the prefrontal cortex (PFC) after 8 weeks of AE, there was a decrease in CHOP, GRP78, p-eIF2α, eIF2α, p-PERK, and PERK expression. However, no significant differences were observed in the levels of ATF4, p-IRE1, IRE1, and p-JNK. In the cortex, the low capacity run group (LR) displayed no differences in the ATF6, eIF2α, GRP78, and XBP1 expression levels after 3 weeks of AE. In contrast, the high capacity run group (HR) displayed increased ATF6 and eIF2α expression levels while displaying no changes in the GRP78 and XBP1 levels. In the hippocampus, AE increased the mRNA levels of GRP78, which was restricted to the LR group. No differences were observed in ATF6, eIF2α, and XBP1s. In the HR group, increased mRNA levels of ATF6, eIF2α, GRP78, and XBP1s were observed. When evaluating the gene expressions of ATF6, GRP78, and XBP1, there was a significant increase in LR, however, there were no differences in the eIF2α levels. However, in HR, there was an increase in the expressions of all ATF6, eIF2α, GRP78, and XBP1 indicators. To verify whether changes in the mRNA levels would be accompanied by changes at protein levels, protein expression by Western blotting was performed in the two brain areas. In the hippocampus, there were no differences in the protein levels of ATF6, eIF2a, GRP78, p-PERK, PERK, and XBP1u in both the LR and HR groups. However, in LR, there was an increase restricted to the XBP1s protein levels. Additionally, in HR, there was an increase in p-eIF2α protein levels (Table 4).

In five studies, t20 indicators of ER stress were evaluated, such as ATF3, ATF4, ATF6, BIP, CHOP, p-eIF2α, eIF2α, GRP75, GRP78, GRP94, HSP25, HSP60, HSP70, HSP90, HSC70, p-IRE1, IRE1, p-PERK, PERK, and XBP1 in different types of skeletal muscle. Four studies evaluated the soleus [33–36], two studies evaluated the EDL [34,35], one study evaluated the quadriceps [37], one study evaluated gastrocnemius [36] and one study evaluated the tibialis anterior [33] (Table 3). Firstly, in the soleus muscle, there was a significant reduction in ATF6, BIP, CHOP, eIF2α, GRP78, p-IRE1, and IRE1 after AE intervention. In addition, there was a decrease in ATF6 after a treadmill inclined downwards. However, there were higher levels of ATF6, p-eIF2α, eIF2α, HSP70, HSC70, p-PERK, and PERK after AE, with the absence of inclination. Exposure to 8 weeks of AE on a treadmill with an up and down inclination significantly increased the expression of BIP, p-eIF2α, eIF2α, p-PERK, PERK, p-IRE1, and IRE1. Moreover, there were similar increases when mice were provided 10 weeks of this same protocol with AE. The protein levels, such as ATF6, BIP, p-eIF2α, eIF2α, p-PERK, PERK, p-IRE1, and IRE1, were identified. In this sense, ER stress was associated with pathological conditions. In contrast, there were no significant differences in the levels of HSP25, HSP60, HSP90, GRP75, GRP78, and GRP94 after AE without the inclination protocol. On an 8-week uphill treadmill, there were no differences in ATF6. The same was observed in a 10-week uphill treadmill, with no differences in the ATF6, BIP, p-IRE1, and IRE1 expression levels.

**Table 3.** Sample characteristics and different physical exercise protocols.

| Author, Year | Species | Sex | Age/*n* | Intervention | Physical Exercise Protocol Characteristics |
|---|---|---|---|---|---|
| Belaya, 2018 | ICR mice | Male | 3 months old/*n* = 22 | Aerobic Exercise | F: Not applicable; D: 21 weeks/free movement on running wheel; I: not applicable |
| Chang, 2020 | C57BL/6J mice | Male | 4–20 months old/*n* = 64 | Swimming | F: 5 days/week; D: 8 weeks/60 min per session; I: not applicable (free of any loading) |
| Feng, Li 2018 | Sprague Dawley rats | Male | 7 weeks old/*n* = 150 | Aerobic Exercise | F: 5 days/week; D: 8 weeks/40 min per session; I: 18 m/min of speed |
| Kim, 2010 | C57BL/6J mice | Male | 8 weeks old/*n* = 20 | Aerobic Exercise | F: Not applicable; D: 3 weeks/free movement on running wheel; I: not applicable |
| Kim, 2018 | Sprague Dawley rats | Male | 50 weeks old/*n* = 30 | Aerobic and Resistance Exercise | Aerobic Exercise: F: 3 days/week; D: 12 weeks/30 min per session; I: in the first week, the rats underwent 5 min of exercise at a speed of 10 m/min; after that, the intensity was maintained in 60% of VOmax or 22 m/min. Resistance Exercise: F: 3 days/week; D: 12 weeks/60 min per session; I: eight sets were used with different volume percentages (30%, 40%, 50%, 70%, 80%, 90%, and 100%) of the rat's body weight with 2 min of rest |
| Ma, 2021 | C57BL/6J mice | Male | 8 weeks old/*n* = 12 | Aerobic Exercise | F: 5 days/week; D: 8 weeks/60 min per session; I: 12–15 m/min |
| Murlasits, 2007 | Sprague Dawley rats | Male | 4 months old/*n* = 54 | Aerobic Exercise | F: Not applicable; D: 1 week (short-term)/60 min per session; I: 70% of VOmax |
| Pereira, 2016 | C57BL/6J mice | Male | 8 weeks old/*n* = 48 | Aerobic Exercise | F: 5 days/week; D: 8 weeks/60 min per session; I: first to fifth week (60% of EV), sixth week (70% of EV), seventh week (75% of EV), in the eighth week, two training sessions with 4 h of interval applied. |
| Pinto, 2017 | C57BL/6N mice | Male | 8 weeks old/*n* = 48 | Aerobic Exercise | F: 5 days/week; D: 8 weeks/10 min per session; I: 3 m/min with 2 days of recovery |
| Pinto, 2019 | C57BL/6N mice | Uninformed | 8 weeks old/*n* = 15 | Aerobic Exercise | F: Not applicable; D: 1 week (acute exercise)/60–90 min; I: 22 m/min |
| Vicente, 2020 | C57BL/6N mice | Uninformed | 8 weeks old/*n* = 40 | Aerobic Exercise | F: Uninformed; D: Uninformed; I: 45–60% of EV |

**Table 3.** *Cont.*

| Author, Year | Species | Sex | Age/*n* | Intervention | Physical Exercise Protocol Characteristics |
|---|---|---|---|---|---|
| Vicente, 2021 | C57BL/6N mice | Male | 8 weeks old/*n* = 40 | Aerobic Exercise | F: 5 days/week; D: 4 weeks/60 min per session; I: in the adaptation, a speed of 6 m/min was used. A total of 60% of the EV was used every week, with a 48-h interval between sessions. |
| Wu, 2011 | C57BL/6N mice | Male/Female | Uninformed | Aerobic Exercise | F: 5 days/week; D: 4 weeks/10, 15, 30, 45, and 60 min per session; I: three speed standards were used: 5 m/min (warm-up), 20 m/min, and 25 m/min until exhaustion |
| Yang, 2015 | Sprague Dawley rats | Male | 5 weeks old/*n* = 24 | Aerobic Exercise | F: 5 days/week; D: 8 weeks/60 min per session; I: 26 m/min of speed |

Caption: D: Duration; EV: Exhaustion Velocity; I: Intensity; F: Frequency; m: meters; min: minutes; *n*: total number of rodents VO2max: Maximum Oxygen Volume.

**Table 4.** Impacts of different physical exercise protocols on ER stress indicators in central and peripheral tissues.

| | Tissue | Endoplasmic Reticulum Stress Markers after Exercise |
|---|---|---|
| Belaya et al., 2018 | Skeletal Muscle | Soleus: = HSP25, HSP60, HSP90, GRP75, GRP78; ↑ HSP70, HSC70; ↓ CHOP Anterior Tibial: = HSP25, HSP60, HSP70, HSP90, HSC70, GRP75, GRP78, CHOP; ↑ GRP75 |
| Chang et al., 2020 | Heart | Myocardium: ↑ PDE5; ↓ CHOP, GRP78, PERK, PKG; |
| Feng, Li et al., 2018 | Prefrontal Cortex | =ATF4, p-IRE1, IRE1, p-JNK; ↓ CHOP, GRP78, p-eIF2α; eIF2α; p-PERK/PERK |
| Kim et al., 2010 | Cortex, Hypothalamus, and Hippocampus | LR Cortex (mRNA): = ATF6, eIF2α; GRP78, XBP1. HR Cortex (mRNA): ↑ ATF-6, eIF2α, GRP78, XBP1 LR Hippocampus (mRNA): = ATF6, eIF2α, XBP1/↑ GRP78. (Protein): =ATF-6, p-eIF2α, eIF2α, GRP78, p-PERK/PERK, XBP1u; ↑ XBP1s. HR Hippocampus (mRNA): ↑ ATF6, eIF2α, GRP78, XBP1. (Protein): = ATF6, eIF2α, GRP78, p-PERK/PERK, XBP1s, XBP1u; ↑ p-eIF2α LR Hypothalamus (mRNA): = eIF2α; ↑ ATF6, GRP78, XBP1. HR Hypothalamus (mRNA): ↑ ATF-6, eIF2α, GRP78, XBP1 |
| Kim et al., 2018 | Heart | Myocardium: AE: = GRP78; ↓ CHOP, p-PERK/PERK RE: = CHOP, GRP78;↓ p-PERK/PERK; |
| Ma et al., 2021 | Heart | Myocardium: ↓ ATF4, CHOP, p-eIF2α, eIF2α, GRP78 |
| Murlasits et al., 2007 | Heart | Myocardium: = ATF3, CHOP, GRP94; ↑ HSP72; ↓ GRP78 |

**Table 4.** *Cont.*

| | Tissue | Endoplasmic Reticulum Stress Markers after Exercise |
|---|---|---|
| Pereira et al., 2016 | Skeletal Muscle | OTR EDL (8 Wk): = ATF6, BIP, p-eIF2$\alpha$, eIF2$\alpha$ p-IRE1, IRE1, p-PERK/PERK. EDL (10 Wk): ↑ p-PERK/PERK;↓ ATF6;↓ BIP; = p-eIF2$\alpha$, eIF2$\alpha$, p-IRE1, IRE1 OTR-U EDL (8 Wk): = ATF6, p-eIF2$\alpha$, eIF2$\alpha$, p-IRE1, IRE1; ↑ p-PERK/PERK; ↓ BIP. EDL (10 Wk): = ATF6, p-IRE1, IRE1, p-PERK/PERK; ↑ BIP, p-eIF2$\alpha$, eIF2$\alpha$ OTR-D EDL (8 Wk): ↑ ATF6, BIP, p-eIF2$\alpha$; eIF2$\alpha$; p-IRE1, IRE1, p-PERK/PERK. EDL (10 Wk): = ATF6, ↑ BIP, p-eIF2$\alpha$, eIF2$\alpha$, p-IRE1, IRE1; p-PERK/PERK OTR Soleus (8 Wk): = BIP;↑ p-eIF2$\alpha$; eIF2$\alpha$; p-PERK/PERK;↓ ATF6, p-IRE1, IRE1. OTR Soleus (10 Wk): ↑ ATF-6; = p-PERK/PERK; ↓ BIP, p-IRE1, IRE1 OTR-U Soleus (8 Wk): = ATF6; ↑ BIP, p-eIF2$\alpha$; eIF2$\alpha$; p-PERK/PERK; p-IRE1, IRE1. OTR-U Soleus (10 Wk): = ATF-6, BIP, p-IRE1, IRE1; ↑ p-eIF2$\alpha$; eIF2$\alpha$; p-PERK/PERK OTR-D Soleus (8 Wk): ↑ BIP, p-eIF2$\alpha$; eIF2$\alpha$; p-IRE1, IRE1, p-PERK/PERK; ↓ ATF6. OTR-D Soleus (10 Wk): ↑ ATF-6, BIP, p-eIF2$\alpha$; eIF2$\alpha$; p-IRE1, IRE1, p-PERK/PERK |
| Pinto et al., 2017 | Hypothalamus | OTR and OTR-U: = ATF-6; ↑ BIP, GRP94, p-IRE1, IRE1; OTR-D: ↑ ATF-6, BIP, GRP94, p-IRE1, IRE1 |
| Pinto et al., 2019 | Skeletal Muscle | EDL: = ATF6, p-eIF2$\alpha$; eIF2$\alpha$; ↑ BIP, CHOP. Soleus: ↓ ATF-6, BIP, CHOP, eIF2$\alpha$; ↑ p-eIF2$\alpha$ |
| Vicente et al., 2020 | Heart | Left Ventricle: ↑ BIP, p-PERK/PERK |
| Vicente et al., 2021 | Heart | Left Ventricle: = CHOP, p-eIF2$\alpha$; eIF2$\alpha$; JNK; ↓ p-IRE1, IRE1 |
| Wu et al., 2011 | Skeletal Muscle | Quadriceps: = GRP94, XBP1t; ↑ ATF3, ATF-4, BIP, CHOP, XBP1s |
| Yang et al., 2015 | Skeletal Muscle | Gastrocnemius and Soleus: = GRP94; ↓ CHOP, GRP78 |

Caption: ATF3: Activating Transcription Factor 3; ATF4: Activating Transcription Factor 4; ATF6: Activating Transcription Factor-6; BIP: Binding Protein; CHOP: C/EBP Homologous Protein; EDL: Extensor Digitorum Longus; p-eIF2$\alpha$: Eukaryotic Translation Initiation Factor 2$\alpha$ Phosphorylated; eIF2$\alpha$: Eukaryotic Translation Initiation Factor 2$\alpha$; p-IRE1: Inositol-Requiring Enzyme 1 phosphorylated; IRE1: Inositol-Requiring Enzyme 1; JNK: c-Jun N-Terminal Kinase; GRP75: Glucose-regulated protein 75; GRP78: Glucose-regulated protein 78; GRP94: Glucose-regulated protein 94; HR: High-Runner; HSC70: Heat Shock Cognate 70; HSP25: Heat Shock Protein 25; HSP60: Heat Shock Protein 60; HSP70: Heat Shock Protein 70; HSP90: Heat Shock Protein 90; LR: Low-Runner; OTR: Overtrained by running without inclination; OTR-D: Overtrained by downhill running; OTR-U: Overtrained by uphill running; PDE5: Phosphodiesterase 5A; PKG: Protein Kinase CGMP; p-PERK: PKR-like ER phosphorylated kinase; PERK: PKR-like ER kinase; XBP1: X-Box Binding Protein-1; XBP1s: X-Box Binding Protein-1 Encoding Spliced; XBP1u: X-Box Binding Protein Unspliced; XBP1t: X-Box Binding Protein Total.

In the EDL muscle, there was a decrease after 10 weeks of AE in ATF6 and BIP. However, there was an increase in the levels of p-PERK and PERK. When acute AE was applied for 1 week, there were higher levels of BIP and CHOP. After 8 weeks, AE with up and down inclination led to an increase in p-PERK and PERK expression levels. Furthermore, the AE protocol, with the treadmill inclination downward only, was responsible for the increase in ATF6, BIP, p-eIF2$\alpha$, eIF2$\alpha$, p-IRE1, and IRE1. Similarly, after 10 weeks of upward inclination, AE was able to increase BIP, p-eIF2$\alpha$, and eIF2$\alpha$. Then, in the AE protocol with the treadmill incline down, high levels of BIP, p-eIF2$\alpha$, eIF2$\alpha$, p-IRE1, IRE, p-PERK, and PERK were found. There were no differences in the levels of ATF6 (1 and 8 weeks) and p-eIF2$\alpha$/eIF2$\alpha$ (1, 8, and 10 weeks); additionally, increases were also found in p-PERK/PERK (8 weeks), p-IRE1, and IRE1 (10 weeks) after AE without an inclination. In the upward-inclination AE after 10 weeks, no significant differences were found in ATF6, p-IRE1, and IRE1. There were no differences after 8 weeks of upward-inclination AE in ATF6, p-eIF2$\alpha$, eIF2$\alpha$, p-IRE1, and IRE1. Finally, after 10 weeks with a downward inclination, only ATF6 showed no significant differences.

In the quadriceps after AE for 4 weeks, there was an increase in ATF3, ATF4, BIP, CHOP, and XBP1s. However, there were no significant differences in the GRP94 and XBP1t expression levels. In the gastrocnemius muscle, there was a decrease in the CHOP and GRP78 expression levels after 8 weeks of AE. No statistical differences were observed in the GRP94 levels. Finally, in the tibial anterior muscle, an increase in GRP75 was only found after 21 weeks of AE. However, at the levels of CHOP, GRP78, HSP25, HSP60, HSP70, HSP90, and HSC70, no differences were found (Table 4).

*Methodological Quality of Studies*

The items were evaluated individually using the SYRCLE instrument. It was observed that all included studies made it clear how the allocation sequence was performed. In addition, the groups acted in accordance with the experimental protocol in all studies. Additionally, the allocation of animals and housing was performed equally. Due to the structure of the different AE protocols, the rodents of each group were randomly selected. However, none of the studies performed outcome selections nor presented a high risk of bias. All studies were presented at completion with a score of 8 (Table 5).

**Table 5.** Methodological quality of studies using the SYRCLE strategy.

| Author, Year | Q1 | Q2 | Q3 | Q4 | Q5 | Q6 | Q7 | Q8 | Q9 | Q10 | Score |
|---|---|---|---|---|---|---|---|---|---|---|---|
| Belaya et al., 2018 | Y | Y | Y | Y | N | Y | N | Y | Y | Y | 8 |
| Chang et al., 2020 | Y | Y | Y | Y | N | Y | N | Y | Y | Y | 8 |
| Feng, Li et al., 2018 | Y | Y | Y | Y | N | Y | N | Y | Y | Y | 8 |
| Kim et al., 2010 | Y | Y | Y | Y | N | Y | N | Y | Y | Y | 8 |
| Kim et al., 2018 | Y | Y | Y | Y | N | Y | N | Y | Y | Y | 8 |
| Ma et al., 2021 | Y | Y | Y | Y | N | Y | N | Y | Y | Y | 8 |
| Murlasits et al., 2007 | Y | Y | Y | Y | N | Y | N | Y | Y | Y | 8 |
| Pereira et al., 2016 | Y | Y | Y | Y | N | Y | N | Y | Y | Y | 8 |
| Pinto et al., 2016 | Y | Y | Y | Y | N | Y | N | Y | Y | Y | 8 |
| Pinto et al., 2019 | Y | Y | Y | Y | N | Y | N | Y | Y | Y | 8 |
| Vicente et al., 2020 | Y | Y | Y | Y | N | Y | N | Y | Y | Y | 8 |
| Vicente et al., 2021 | Y | Y | Y | Y | N | Y | N | Y | Y | Y | 8 |
| Wu et al., 20211 | Y | Y | Y | Y | N | Y | N | Y | Y | Y | 8 |
| Yang et al., 2014 | Y | Y | Y | Y | N | Y | N | Y | Y | Y | 8 |

## 4. Discussion

The purpose of the present systematic review was to synthesize the scientific literature on the role of PE on molecular markers of ER stress in the central and peripheral tissues of rodents. Our results demonstrated that different PE protocols were able to reduce the levels of expression of molecular and cellular signals of ER stress in cardiac, muscular, and brain tissues.

Several studies demonstrate that exposure of cardiac tissue to ER stress appears to trigger oxidative stress, ischemia, disturbances in calcium metabolism, and diseases, including systemic arterial hypertension, myocardial infarction, and heart failure [38,39]. In this sense, the investigation of ER cardiac stress markers has been used in the possible identification of therapeutic targets since they are scarce; therefore, interventions with factors related to lifestyle, with PE being among them, are essential. The results demonstrate that AE, swimming, and RE were effective in reducing the levels of ER stress signals in cardiac tissue. Hong et al. [40] point out that a decrease in the expression of factors, such as ATF4, BIP/GRP78, p-PERK, and PERK, improves cardiac function and reduces the risk of infarction. Furthermore, the control of inflammatory processes and apoptosis are also related to these benefits [41,42].

The brain is anatomically divided into areas, including the hypothalamus, prefrontal cortex, cortex, and hippocampus, which are important in motor, sensory, food intake, and cognitive control [43–46]. Moreover, it has a high phenotypic plasticity rate, especially in critical periods of development, making it more susceptible to stressful stimuli. Evidence observed that these stimuli can promote the overactivation of UPR and its molecular signals promoting dysfunctions, and neurodegeneration [17,47]. Among these regions, the hypothalamus is responsible for integrating neural, nutritional, and hormonal signals, as well as the circadian rhythm, body temperature, thirst, food intake, energy expenditure, and glucose metabolism, effectively contributing to the maintenance of body homeostasis [48,49].

However, changes, such as an excessive intake of caloric and high-fat foods, metabolism, and inflammation lead to the activation of ER stress signals in the hypothalamus, contributing to being overweight and obese [50]. In contrast, it is widely known that PE plays an important role in the fight against obesity and being overweight [51,52]. Surprisingly, the results of this systematic review observed higher expressions of ATF6, BIP, GRP94, p-IRE1, and IRE1 after 8 weeks of AE with different inclinations. It is generally seen that PE protocols on a treadmill inclined up or down can promote excessive levels of metabolic stress, inflammation, and cellular apoptosis in rodents, being responsible for the activation of signaling pathways to ER stress [53]. Furthermore, Kim et al. [32], using voluntary physical exercise on a running wheel, observed an increase in the expression of ATF6, GRP78, and XBP1 in the low-capacity run (LR) group, and ATF6, eIF2$\alpha$, GRP78, and XBP1 in the high capacity run mice (HR). Altogether, these results suggest that both a reduction and excess of PE can impair cellular homeostasis in the hypothalamus.

The regions of the cortex and PFC are fundamental in motor control and development, as well as the planning of complex behaviors and thoughts, personality expression, decision-making, and behavior modulation [54,55]. The PE showed different responses in the ER stress indicators in these regions. In the cortex, the gene expression levels of ATF6, eIF2$\alpha$, GRP78, and XBP1 after 3 weeks of PE were evaluated, with an increase only in ATF6 and eIF2$\alpha$. The effects of chronic exposure to ER stress include a worsening of learning and memory capacity, as well as an increase in cell apoptosis [56,57]. However, in PFC after 8 weeks, PE was able to significantly decrease the levels of CHOP, GRP78, p-eIF2$\alpha$, eIF2$\alpha$, p-PERK, and PERK. Evidence supports these findings, demonstrating that PE positively regulates inhibitory control, cortical volume, and behavior, preventing the onset and advancement of anxiety and depression in rodents and translationally in humans. According to several authors, one of the mechanisms behind these benefits is the control of ER stress [15,41,58].

The hippocampus is a component of the limbic system and acts as a control of emotions and different types of memory. The results identified an increase in ATF6, eIF2$\alpha$, GRP78, and XBP1s after 3 weeks of AE. However, the evidence points in another direction, demonstrating that intervention, mainly with AE, promotes better emotional control, acting in a preventive way against psychiatric disorders, including anxiety and depression [59–61]. Furthermore, the role of this modality is highlighted in cognitive improvement, particularly in memory capacity, under both normal and pathological conditions. In contrast, when PE is applied exhaustively, it can activate the signaling pathways related to cell damage, and

it is necessary to develop a PE protocol that considers biological specificity and individuality, thus avoiding possible harm to the brain and organism homeostasis [62–64]. The variability in the expression of ER stress markers in different brain regions may be related to the variability in age (7–50 weeks) of the animals exposed to different training protocols. Evidence, as pointed out in their review study, is that age is a determining factor for the development or protectiveness of the organism from ER stress. The authors argue that this occurs mainly due to the reduction in the functionality of the UPR recognition systems and to stress, culminating in drastic consequences for cell homeostasis [62,64].

Skeletal muscle has a wide variety of functions, including voluntary contraction, metabolic control, secretion of hormones, and myokines [12,65–67]. The summary of the evidence included in the present review indicates a diversity (i.e., downregulation or upregulation) in the ER stress markers caused by the AE without the use of inclination. The ability of AE to protect the cell against different types of stress is widely known. Among them is oxidative stress (OS), which is related to excess reactive oxygen species (ROS) or nitrogen, as well as a reduction in antioxidant activity and damage to cellular components, including the plasma membrane [68–70]. Additionally, AE, mainly at a moderate intensity, can potentiate the action of antioxidant defenses, preventing the establishment of OS. Dandekar et al. demonstrate that one of those responsible for the activation of ER stress is EO, which occurs mainly due to the connectivity between the ER and mitochondria, which is a major site of ROS production and is related to pathological conditions [71,72]. Therefore, the AE plays a fundamental role in the regulation of these processes when properly structured. However, when the AE promotes high demands of muscular work, leading to exhaustion and voluntary motor failure conditions, for example, by the inclusion of inclination, they lead to severe metabolic disorders, including ER stress. The included findings confirm this perspective, given there was a significant increase in the ER stress markers in different muscle structures after the application of the AE on a motorized treadmill inclined up or down. The evidence suggests that this type of PE should be avoided in order to contain the activation of the markers linked to cellular stress. Another point for consideration in the activation of ER stress markers is the morphological and physiological differences of each tissue evaluated in the present systematic review. It is known that each of these tissues responds differently to physical exercise and its protocols; among these main adaptations that can explain these divergences are the metabolic ones, since each tissue prioritizes the use of energy substrates, directing the activation of aerobic or anaerobic reactions and fiber-type predominance in, for example, skeletal muscle. Finally, the present study also makes clear the need for the evaluation, planning, and prescription of each physical exercise protocol, while considering the stimuli, responses, and physiological adaptations that can be produced in the body in peripheral and central tissues. For this, the variables related to physical exercise or training, including volume, duration, frequency, the type of aerobic or anaerobic exercise, and mainly the intensity and inclination of each protocol, are capable of positively modulating the signaling pathways and expression of markers of the ER stress, preventing the deregulation of cellular homeostasis and the emergence of chronic diseases.

This review was structured in several points: (I) This is the first systematic review to evaluate the impacts of PE on cellular and molecular indicators of ER stress in the heart, brain, and skeletal muscle in preclinical studies with rodents. (II) We systematically searched the available literature to evaluate the different PE protocols, including aerobic, swimming, and resistance exercises, considering the different effects caused by each modality on the evaluated parameters of ER stress. (III) Twenty-five different markers of ER stress were found in the included studies—to demonstrate the complexity and importance of this process—at the mechanisms and application levels. Moreover, there were different ways of evaluating these markers via the mRNA and protein expression levels. (IV) The current work presents all the characteristics related to the variables that make up the intervention with PE, including the type/modality, frequency, protocol, and session duration and intensity. (VI) Our study can be considered a theoretical basis for establishing future exercise

prescription guidelines to combat clinical conditions and diseases that have ER stress as part of their etiology and pathophysiology.

Among the limitations, it is known that preclinical studies are used to substantiate knowledge about the mechanisms and responses in less evolved organisms, and this attributes little clinical applicability and external validity to the findings of the present review. However, this study can guide studies and protocols in humans, as well as present the importance of the regular practice of PE and the identification of possible therapeutic targets for the elaboration of drugs and to combat chronic diseases related to ER stress.

## 5. Conclusions

This systematic review demonstrates that exposure to different PE protocols can reduce the activation of ER stress markers in different areas. In cardiac tissue, a reduction in ATF4, CHOP, GRP78, p-PERK, PERK, and PKG was observed. Among the cortical regions evaluated, the CPF was the one that showed more responses to reduce the RE indicators (CHOP, GRP78, p-eIF2$\alpha$, eIF2$\alpha$, p-PERK, and PERK) after PE. Furthermore, the soleus and EDL skeletal muscles showed reductions in markers, such as ATF6, BIP, CHOP, eIF2$\alpha$, GRP78, p-IRE1, and IRE1. However, in the quadriceps and gastrocnemius, no reductions were observed in these markers. In addition, exhaustive and inclination protocols have shown effects contrary to the reduction of ER stress; therefore, they should be avoided to prevent cellular stress that acts as a trigger of numerous chronic diseases.

**Author Contributions:** M.S.d.S.F. and C.J.L. conceived the study idea and design; M.S.d.S.F., G.C.J.S., F.JA., R.d.S.H., T.O.F., F.O.S. and C.J.L. performed the search in databases; M.S.d.S.F., G.C.J.S. and T.O.F. conducted data extraction; M.S.d.S.F., G.B. and G.C.J.S. performed the methodological quality analysis; M.S.d.S.F., R.F.d.S., F.J.A., P.C.B., G.B. and C.J.L. wrote the manuscript with the review, editing, and final approval from all authors. All authors have read and agreed to the published version of the manuscript.

**Funding:** This research received no external funding.

**Institutional Review Board Statement:** Not applicable.

**Informed Consent Statement:** Not applicable.

**Data Availability Statement:** The data used to support the findings of this study are included within the article.

**Acknowledgments:** The authors are thankful to FACEPE and CNPq (Foundation for the Support of Science and Research from Pernambuco State—Brazil, APQ-0765-4.05/10; -1026-4.09/12; Universal-408403/2016) for the financial support provided to acquire the equipment and reagents used in this work. This study was financed in part by the Coordenação de Aperfeiçoamento de Pessoal de Nível Superior—Brasil (CAPES)—Finance Code 001.

**Conflicts of Interest:** The authors declare no conflict of interest.

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
