# Peer review of "Physical Exercise Decreases Endoplasmic Reticulum Stress in Central and Peripheral Tissues of Rodents: A Systematic Review"

_ejihpe, doi:10.3390/ejihpe13060082_

Round 1

Reviewer 1 Report

The review “Physical Exercise Decreases Endoplasmic Reticulum Stress in Central and Peripheral Tissues of Rodents: A Systematic Review” is a very interesting synthesis work dealing with a very important subject, but major information is missing to conclude.

The idea is good but we have many parameters for comparison and the authors didn’t consider them in the discussion.

- The age of the rodents plays a very important role in physiological change and homeostasis metabolism but the author didn’t speak about this.

- The Physical Exercise Protocol Characteristics play a very important role in physiological change and homeostasis metabolism but the author didn’t speak about this.

- The difference in metabolic responses of tissues is very important to discuss, especially when we have the same markers and different results between two tissues, but the author did not talk about it.

- Table 4 needs to be reviewed and simplified to better understand it and to be able to convey clear information.

All these remarks must be taken into consideration. Answers to these criticisms will reinforce the quality of the manuscript and will permit us to conclude with more accuracy.

The author has to show the major revisions in the text, with a different color text, by highlighting the changes.

Author Response

Dear Reviewer, 

I am attaching the answers to the comments.

Best regards, 

Reviewer 2 Report

In this study, Santos de Sousa Fernandez et al studied the stress-induced damage to theEndoplasmic reticulum. Many tissues are affected by Endoplasmic reticulum stress (ER stress), by increasing oxidative stress, ischemia, disturbances in calcium metabolism, and diseases including systemic arterial hypertension, myocardial infarction, and heart failure. Many other tissues are involved. 

Table 4 collects and shows data. The results demonstrate that anaerobic exercise, swimming, and RE were effective in reducing the levels of ER stress signals in cardiac tissue. CNS data of ER stress are ambiguous. 

I have few suggestions and/or questions. Do you think there is an active role of autophagy in this mechanism? A protective role?

How do you explain CNS data? ER stress can be demonstrated or not? 

Thank you, have a good job.

Author Response

(The authors gave the same response as above.)

Round 2

Reviewer 1 Report

In this version of the review “Physical Exercise Decreases Endoplasmic Reticulum Stress in Central and Peripheral Tissues of Rodents: A Systematic Review”  We can see an evolution compared to the first version, especially in the discussion part, because they have become richer, with more explanation.

the authors have considered the reviewer's remarks and suggestions, which has positively impacted the quality and consistency of the review.

the review is accepted for me with this version